# *Quercus petraea* (Matt.) Liebl. from the Thayatal National Park in Austria: Selection of Potentially Drought-Tolerant Phenotypes

Marcela van Loo [1,*], Roman Ufimov [1], Michael Grabner [2], Christian Übl [3], Andrea Watzinger [4], Florian Irauschek [1], Heino Konrad [5], Soňa Píšová [5] and Carlos Trujillo-Moya [1]

1 Department of Forest Growth, Silviculture and Genetics, Austrian Research Centre for Forests (BFW), Seckendorff-Gudent-Weg 8, 1131 Vienna, Austria; roman.ufimov@bfw.gv.at (R.U.); florian.irauschek@bfw.gv.at (F.I.); carlos.trujillo-moya@bfw.gv.at (C.T.-M.)

2 Institute of Wood Technology and Renewable Resources, University of Natural Resources and Life Sciences, Konrad-Lorenz-Strasse 24, 3430 Tulln an der Donau, Austria; michael.grabner@boku.ac.at

3 Thayatal National Park Administration, National Park House, Merkersdorf 90, 2082 Hardegg, Austria; christian.uebl@np-thayatal.at

4 Institute of Soil Research, Department of Forest- and Soil Sciences, University of Natural Resources and Life Sciences, Konrad-Lorenz-Strasse 24, 3430 Tulln an der Donau, Austria; andrea.watzinger@boku.ac.at

5 Department of Forest Biodiversity and Nature Conservation, Austrian Research Centre for Forests (BFW), Seckendorff-Gudent-Weg 8, 1131 Vienna, Austria; heino.konrad@bfw.gv.at (H.K.); sona.pisova@bfw.gv.at (S.P.)

* Correspondence: marcela.vanloo@bfw.gv.at

**Abstract:** The increasing demand for climate-adapted seeds and planting material poses a challenge due to the limited availability, particularly for tree species such as oaks. National parks, known for their large-standing diversity and a wide range of habitats, can serve as valuable sources for identifying trees suitable for both the initiation of tree breeding and conservation strategies. This study aimed to identify valuable forest genetic resources of the Thayatal National Park in Austria by selecting potentially drought-tolerant phenotypes. For this purpose, we selected 404 mature trees of *Quercus petraea* (Matt.) Liebl. from eight populations growing on medium to dry sites in eight populations. Further, we characterized them for autochthony, genetic structure, genetic diversity using genetic markers (plastid- and nuclear-SSRs) and estimated their age. Finally, we applied wood core analysis to estimate tree response to historical drought events to identify the possible drought-tolerant phenotypes. The age of the trees ranged from 29 to 245 years (as of the year 2023). All *Q. petraea* trees were inhabiting a plastid haplotype 17a, autochthonous for this area. Nevertheless, the genetic structure estimated by ten nuSSRs revealed a pronounced structure in the dataset, largely caused by young trees exhibiting lower genetic diversity. A total of 85 elite potentially drought-tolerant trees were finally selected based on their morphological response (resistance, recovery ability, resilience, and relative resilience) to three historical drought events (1992–1994, 1947, 1917). The intrinsic water use efficiency and its difference (iWUE and DWiWUE), estimated by isotope analysis of $\delta 13C$ of latewood in wet (1987) and dry (1994) years, did not correlate with any of the drought response traits (*Rt*, *Rc*, *Rs*, r*Rs*). We discuss the further use of the selected oak trees for the establishment of seed stands and orchards to enhance seed production and the integration of other omics approaches, such as large-scale high-throughput plant phenotyping (HTPP) and transcriptomics, for in-depth analyses of drought tolerance of selected phenotypes.

**Keywords:** sessile oak; genetic structure; genetic diversity; autochthony; wood cores; DNA markers; water use efficiency

## 1. Introduction

Numerous studies have focused on the notable resilience of European oak species to climate change [1–3]. This resilience is particularly evident when the oaks grow in mixed stands with other deciduous tree species [4–6]. Both young seedlings and mature oak

trees display adaptive potential for coping with climate change: Oaks respond flexibly to increased temperatures and drought conditions, rapidly resuming metabolic functions even after prolonged stress periods [7]. Notably, the stomata on the abaxial surface of the leaves remain open during episodes of warming and drought, enabling substantial rates of photosynthesis [8]. Their robust foliage ensures the continuous process of photosynthesis, as well as nutrient assimilation and water translocation. Furthermore, an adjustment in the shoot-to-root growth ratio aids in maintaining an essential water supply. Specifically, during dry spells, there is a proportionally increased allocation of energy directed to root development, which aids water uptake [9–11].

Among the four native oak species (*Quercus cerris* L., *Q. petraea* (Matt.) Liebl., *Q. pubescens* Willd., *Q. robur* L.) distributed in Central Europe [12], only two species, namely *Q. robur* and *Q. petraea* (sessile oak), are of significant importance in forestry. For the latter species, several subspecies, including *Q. petraea* subsp. *austrotyrrhenica* Brullo, Guarino and Siracusa, *Q. petraea* subsp. *huguetiana* Franco and G. López, *Q. petraea* subsp. *pinnatiloba* (K. Koch) Menitsky, and *Q. petraea* subsp. *polycarpa*, are recognized [12]. A recent study focusing on *Q. petraea* has revealed a remarkable phenomenon: rapid evolution and adaptation to climatic changes occurring within just a few generations. This adaptation is facilitated through the enrichment of specific genomic regions with genes associated with plant responses to both pathogens and abiotic stresses, such as temperature fluctuations and drought [13]. These findings underscore the importance of accelerating generational shifts and fostering natural forest regeneration to facilitate rapid stand evolution. Nevertheless, considering current climatic projections, suitable growing locations for tree species are expected to shift. Consequently, the need for artificial planting becomes more pronounced. These imperative interventions align with the goals of the European Green Deal and the EU biodiversity strategy for 2030, which mandate the planting of at least three billion additional trees, strictly adhering to ecological principles [14]. This initiative highlights the critical importance of using climate-adapted forest reproductive material in achieving these goals.

The growing demand for climate-adapted forest reproductive material (FRM), which includes seeds and planting materials, has been well-documented [15,16]. In Austria, most FRM is sourced from harvested seed stands, while seed orchards make a smaller contribution. For key species, such as sessile oak, the majority of FRM (60%) is imported (Ch. Wurzer, Federal Forest Office, pers. comm.). This reliance on foreign sources may stem from Austria's limited number of small seed crop stands and their irregular fruiting patterns.

With projected temperature increases of approximately 5 °C by the end of the century under the RCP 8.5 scenario [17], diverse strategies are being employed to adapt forests to climate change. These include the formulation of suitable forest management strategies, such as embracing mixed-species forests [18], assisted migration [19,20], and breeding. Beyond the current emphasis on pathogen resistance in tree breeding [21], there is a directed effort towards breeding drought- and heat-tolerant tree species [22,23]. Tree breeding can exploit natural biodiversity for sustainable management and conservation by selecting phenotypes and genotypes of particular interest. National parks are crucial reservoirs of habitat and genetic diversity. They often house autochthonous populations native to a particular location, of which a longstanding presence in a specific site [24] reflects their adaptation to local conditions.

Within autochthonous white oak populations across Europe, chloroplast DNA (cpDNA) variation distribution demonstrates a distinct geographic pattern [25–28]. The integration of cpDNA variation with pollen fossil records has facilitated the estimation of post-glacial re-colonization dynamics from glacial refugia [26]. White oak refugia were not only restricted to regions located around the Mediterranean and the Black Sea: the Iberian Peninsula, the Apennine Peninsula, South-east Europe, South-west Asia, and North Africa [26,29,30] but also to northerly situated cryptic refugia [31,32]. The *Q. petraea* subsp. *austrotyrrhenica*, *Q. petraea* subsp. *huguetiana*, and *Q. petraea* subsp. *pinnatiloba* (K. Koch) Menitsky reflect the presence of the *Q. petraea* s.l. refugia around the European Mediterranean arc [12].

While various cpDNA markers, such as cpSNPs [27] and cpSSRs [33,34], have been utilized to explore cpDNA polymorphism, each with its unique nomenclature for cpDNA haplotypes, none have achieved extensive coverage across Europe or revealed the depth of haplotype diversity, as seen in studies conducted primarily two decades ago using the PCR-RFLP method [25,26,35–38]. It has, therefore, been suggested that haplotype distribution maps can help to distinguish introduced from autochthonous populations and provide a framework for identifying the geographical origin of seed lots [38]. Employing this method, a total of six different haplotypes originating from refugia in Italy and the Balkans were previously identified in Austria [25,39], the country of the current study, which was a crossroads for the post-glacial recolonization of oak.

Plant adaptation to drought is a multifaceted phenomenon that can be explored through anatomical/morphological, physiological/biochemical, and genetic lenses [23,40]. Morphological investigations often center on xylem morphology and tree-ring measurements, with dendroecological assessments using annual increments. When combined with meteorological data, these enable the assessment of tree resilience to climate extremes through growth-based resilience indices [41,42]. Physiological and biochemical inquiries focus on traits, such as photosynthesis and water use efficiency (WUE). WUE, which is the net photosynthesis to water transpiration ratio, offers insights into water deficit adaptation and is influenced both environmentally and genetically [43]. For instance, pedunculate oak genotypes with distinct WUE phenotypes reveal divergent drought coping strategies, as evidenced by differences in gene expression associated with drought responses and stomatal regulation [44].

The main objective of this study was to identify and select the most promising drought-adapted phenotypes of *Q. petraea* within the designated study area, the Thayatal National Park. This park, located in the Pannonian north-east of Austria, is characterized by extremely dry sites and a history of drought events. To achieve this goal, we followed a structured approach, where we first assessed the autochthony of *Q. petraea* as a sign of the local adaptation. We then evaluated the genetic diversity and the genetic structure, which might not confound the drought response. Finally, we grouped *Q. petraea* trees based on their drought tolerance estimated by five traits derived from wood core analyses, concluding with the selection of the final potentially drought-tolerant phenotypes. Employing different dendroecological and genetic methodologies, we sought to answer three specific questions:

1.  Are the Thayatal National Park's sessile oaks of autochthonous origin?
2.  Is there evidence of genetic structure among sessile oaks across the Thayatal National Park? If so, how is it reflected in patterns of genetic diversity?
3.  Do the sessile oaks in the Thayatal National Park exhibit variations in drought tolerance differences, and if so, how many potentially drought-tolerant phenotypes can be identified among the studied trees?

The findings are discussed in the context of their broader implications for establishing seed stands and orchards to enhance seed production and their integration into large-scale high-throughput plant phenotyping (HTPP) and RNA-Seq for a comprehensive understanding of drought responses.

## 2. Material and Methods

### 2.1. Study Area and Tree Selection

The studied populations of *Q. petraea* are located within the Thayatal National Park in Austria (Figure 1), established in 2000 and named after the meandering Thaya River, forming a shared border with the Czech Republic. Recognized by the IUCN as a Category II protected area in 2001, the Thayatal National Park covers 1360 hectares in Austria. The overall protected area, including the Czech part established in 1991, spans approximately 7700 hectares [45]. The meandering Thaya River has shaped the valley over millions of years, resulting in a diverse range of exposures and geological attributes that influence the spatial distribution of the vegetation [46,47]. The subsoil conditions vary from acidic to

alkaline due to the underlying rock composition (geological substratum). These distinct geological and geomorphological features, coupled with the unique location of the park in the overlapping area between dry Pannonian-continental and already wet Atlantic-influenced climate, contribute to the large habitat diversity, resulting in a high species biodiversity of the park, with over 40% of Austria's plant species and numerous endangered species [45].

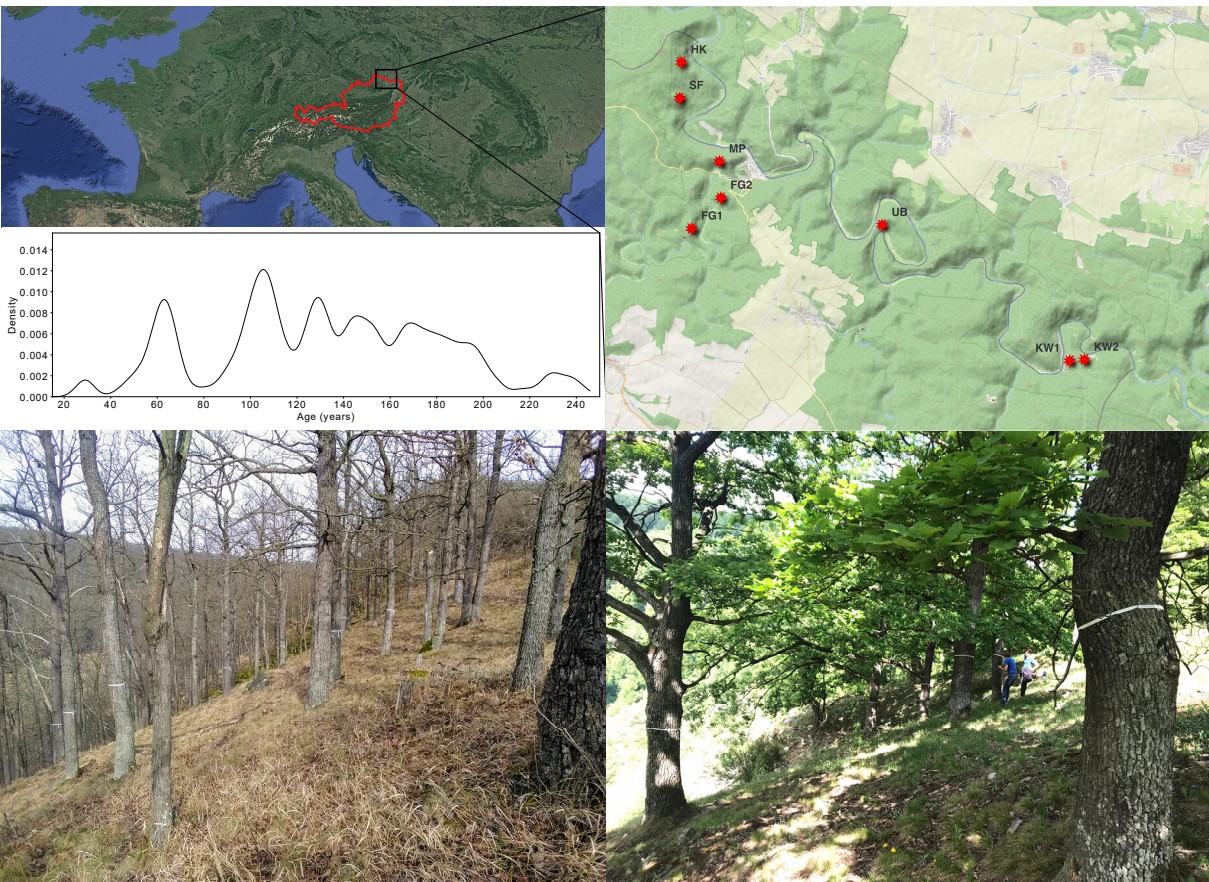

**Figure 1.** An overview map of Europe highlights the location of the Thayatal National Park (black rectangle) in Austria (the border of the country in red line) and eight sites (red stars) where oak samples were collected. Below, photos depict the park's distinctive oak landscape. The accompanying histogram illustrates the age range of the studied oaks in the year 2023.

The National Park is predominantly covered by forests, encompassing more than 90% of its total area [48]. Oak forests predominate in the eastern part of the park on acid granite, while beech forests prevail in the western region over limestone and granite. Coniferous species are less prominent. Human intervention, particularly forestry practices, has left its imprint on the structure of the park, predominantly on accessible plateau sites, where natural deciduous forests were converted into commercially managed forests. These changes have been actively reversed by the forest management of the National Park Administration.

*Quercus petraea* exhibits non-random spatial distribution in the park, forming distinct populations. The park's staff initially selected relevant oak populations, followed by an evaluation conducted by a research team from the Austrian Research Centre for Forests (BFW, Vienna, Austria) to ensure the suitability of the selected trees and populations for study purposes. Sites that exhibited signs of management (e.g., coppicing) or were dominated by other oak species were excluded from the study. Subsequently, in the spring of 2020, old and vital sessile oaks were identified and marked for further investigation.

### 2.2. *Tree Characterization, Material Collection and Processing*

For this study, the location of each individual tree was recorded using GPS coordinates, and the diameter at breast height (DBH) was measured. In 2020, two to three cambium samples were gathered using a hollow punch (2 cm diameter) at the base of the trunk, and two wood cores at breast height were collected with the Haglöf Mora-Coretax increment borer from all selected trees. The cambium samples were used for analyses of autochthony, genetic structure, and diversity. The wood cores were taken for tree ring analysis, allowing us to estimate the age of the trees, tree ring pattern, and drought tolerance. Cambium samples were dried in labeled bags with silica gel and stored until DNA extraction. Two samples had ambiguous labels and were excluded from the genetic analysis. The DNA from 402 trees was extracted using the Qiagen DNeasy 96 Plant Kit (Qiagen, Hilden, Germany) according to the manufacturer's instructions. The wood cores (two per each tree, in total 808 wood cores) were cut with razor blades, scanned, and the tree ring width was measured to an accuracy of 0.001 mm using Windendro 2018 software (https://regentinstruments.com, accessed on 12 September 2023). All samples were synchronized using standard dendrochronological methods (see, for example, [49,50]).

### 2.3. *Genotyping for Analysis of Autochthony, Genetic Diversity and Genetic Structure*

We used ten chloroplast microsatellite (cpSSR) primer pairs selected from previous studies [51,52] to estimate the autochthony and eight nuclear microsatellites (nuSSRs) primer pairs described by [53,54] to evaluate patterns of genetic structure and diversity (Table S1 Supplementary Materials S1). These were amplified by PCR in multiplex reactions using the QIAGEN Type-it Microsatellite PCR Kit (Qiagen, Hilden, Germany). Primer sequences, fluorescent dyes used, multiplex combinations, and information on the DNA laboratory work and capillary fragment analyses are presented in Supplementary Materials S1.

For the assessment of autochthony, we employed reference samples originating from a Europe-wide study by [25,26] on autochthony and post-glacial recolonization of white oaks. These reference samples had previously been analyzed using the PCR-RFLP method. In our study, a total of 57 reference samples from Austria with six known cp-haplotypes (1, 2, 5, 6, 7, 17a, Table S2, Supplementary Materials S3) resulting from PCR-RFLP analyses [25,26] were included for genotyping, accompanied by 32 samples from the National Park, so each Thayatal population was represented by four randomly selected individuals. The set of reference haplotypes (Table S2, Supplementary Materials S3) was selected based on their presence in Austria and neighboring countries (Czech Republic, Slovakia, Germany, Hungary, and Italy) and the availability of their DNA from previous studies [25,26,39].

Standard descriptive genetic diversity measures were computed for the entire set at both the locus and the population level. These included the mean number of alleles ($Na$), inbreeding coefficient ($Gis$), observed heterozygosity ($Ho$), expected standardized heterozygosity ($Hs$) computed with GenoDive 3.06 [55], and for the population level also the allelic richness ($Ar$) using ADZE software [56]. To account for variations in sample size, rarefaction was employed to adjust the latter two measures ($Hs$ and $Ar$), an option offered within the particular software package. The rarefaction size for $Ar$ was 30, corresponding to the minimum number of genotyped loci in the smallest population. Thereafter, we investigated genetic differentiation between populations by calculating pairwise Jost's D [57] using R package diveRsity [58]. In all genetic structure analyses conducted using the STRUCTURE software [59–61], we did not incorporate any artificial grouping of trees into populations. To provide further insight, we performed a Bayesian cluster analysis with 10 replicates of a run, with a burn-in period of 500,000 followed by 500,000 Markov chain Monte Carlo (MCMC) runs under the admixture model with correlated allele frequencies and no linkage of markers [60]. The number of assumed populations (K) was set from one to eight. We used STRUCTURE Harvester to obtain a consensus estimate of the optimal K value [62], and the software CLUMPP [63] was used with a full search to align the results from multiple runs. The population structure and assignment of individuals to their inferred genetic clusters were visualized by the program Distruct [64]. To better

understand the genetic connections among the individuals sampled, pairwise relatedness was calculated using a moments estimator method in PolyRelatedness v. 1.11b [65] and visualized through a coancestry heatmap using heatmap.2 function from the Gplots v.3.1.1 R package [66].

### 2.4. Wood Core Analyses: Age, Drought Tolerance, and Water Use Efficiency

The age of the trees was calculated from the age at breast height as measured on the wood core samples. If the pith of the trees did not exist, the number of missing rings to the pith was estimated using the curvature and the mean ring width of the adjacent tree rings.

We assessed the drought tolerance of individual trees by examining their response to three historical drought events (1992–1994, 1947, 1917). This evaluation was based on four resilience components (traits) of drought response, namely resistance (*Rt*), recovery (*Rc*), resilience (*Rs*), and relative resilience (r*Rs*) [41]. Drought events and reference wet years were determined based on the Standardized Precipitation Evapotranspiration Index (SPEI) [67] (see Figure S1a); values from the Geosphere Austria meteorological station Retz (six months time-window). We aimed to select specific drought events that can be characterized by high severity (i.e., low SPEI values) and by a sufficient period (i.e., 2–3 years) pre- and post-drought with close to normal or wet weather conditions. Details regarding the consideration of dry/wet periods for the analysis are presented in Supplementary Materials S2. Climate data from 1895 to 2020 were used to analyze annual and seasonal precipitation trends and temperature trends (Figure S1b). Only those trees that experienced all three drought events and displayed an absence of genetic structure were considered for the analysis of the response measured in wood cores. This reduced the number of mother trees from 404 to 253. To test the degree of the genetic determination of the four drought response indicators, we made use of the repeated occurrence of drought events in the study area and calculated the proportion of the total variation in drought response that was due to the differences between individuals (the 253 individuals exposed to all three drought events). This proportion is defined as repeatability and is the upper limit of the heritability of the given trait, as repeatability includes genetic and environmental sources of variation, whereas heritability includes only genetic differences among individuals [68]. Linear mixed-effect models were fitted to estimate the adjusted repeatability [69] of resistance (*Rt*), recovery (*Rc*), resilience (*Rs*), and relative resilience (r*Rs*) by using the rptGaussian function from the rptR package [70].

Drought events were fitted as a fixed effect, while tree (grouping factor of interest) and age were fitted as random effects in all models. Traits used as response variables were normally distributed (*Rt*, *Rc*, r*Rs*), except for *Rs*, which was Ln transformed to fit a normal distribution, visually inspected (histogram and Q-Q plot) and tested by the Lilliefors test (Figure S2, Supplementary Materials S2), by using the R packages ggplot2 and nortest, respectively.

These 253 trees were independently ranked according to the values for each index and each drought event (1917, 1947, 1992, 1994). Those that consistently ranked above the median (top 126) for the particular trait within all three drought events were selected and grouped as "good". Conversely, those that consistently ranked below 126 for the particular trait were selected as "bad". Afterward, we performed multiple pairwise *t*-test comparisons and visualization by box plots with *p*-values for both groups (good and bad) with the R packages rstatix, ggplot2, and tidyverse to investigate differences between them.

Intrinsic water use efficiency (iWUE) served as the fifth trait for drought tolerance of mature trees. It is advisable to employ iWUE in combination with other parameters when assessing drought tolerance ([23]; see the reasoning within). For its estimation, measurements of carbon stable isotope ratio ($\delta^{13}C$) in latewood samples of the years 1987 (wet year) and 1994 (dry year) were conducted by continuous flow isotope ratio mass spectrometer coupled to an elemental analyzer (Flash EA—Delta V advantage, Thermo Fisher Scientific, Bremen, Germany). For this purpose, ≤1 mg latewood from the designated years was cut under the microscope and weighed into tin cups. The isotope ratio was expressed as

$\delta^{13}$C values in ‰ normalized against the standard VPDB (Vienna-PDB). The combined uncertainty of measurements (1σ) was 0.15 ‰. The mean standard deviation of replicate tree rings was 1σ = 0.50 ‰ (1994) ($n$ = 85) and 0.58 ‰ (1987). The $^{13}$C discrimination of the plant tissues $\delta^{13}$C, the intercellular $CO_2$ concentration $c_i$, and the iWUE were calculated by the equations [71]:

$$\delta^{13}C = (\delta^{13}C_{atmosphere} - \delta^{13}C_{latewood})/(1 + \delta^{13}C_{latewood}/1000), \tag{1}$$

$$c_i = c_a[(\Delta^{13}C - a)/(b - a)], \tag{2}$$

and

$$iWUE = c_a [1 - (c_i/c_a)]/1.6, \tag{3}$$

with $c_a$ standing for atmospheric $CO_2$ concentration, a—for fractionation during stomatal diffusion, and b—for fractionation during carboxylation. The $CO_2$ concentrations and $\delta^{13}$C values were 348.0 ppmv and −7.70 ‰ and 357.7 ppmv and −7.83‰ for the years 1987 and 1994, respectively. The stomatal diffusion and carboxylation fraction factors were estimated as 4.4 and 25.5 ‰ [71].

For further investigations, we considered three sets of values: iWUE estimated for the wet year, iWUE estimated for the dry year, and their difference, referred to as DWiWUE. In contrast to the analysis of *Rt*, *Rc*, *Rs*, and r*Rs*, the assessment of iWUE and DWiWUE was restricted to only a single pair of years (dry/wet) due to the resource-intensive and time-consuming nature of this approach. Consequently, the robustness of this trait felt short in comparison to the other four traits, and thus, it was tested only if it could provide additional support for the groups of "good" and "bad" phenotypes of the four traits. For their multiple pairwise *t*-test, comparisons and visualization by box plots with *p*-values were implemented in the R packages rstatix, ggplot2, and tidyverse. Furthermore, we extended similar analyses to include DWiWUE.

Finally, the R package corrplot was used to plot a correlogram based on the correlation matrix (Spearman) of all five variables studied, including the age of the 253 trees exposed to all three drought events selected (1917, 1947, 1992–1994). The correlation matrix was ordered according to the degree of association between variables (hierarchical clustering), and the significance threshold was set at a *p*-value of 0.01.

## 3. Results

In total, 404 individual *Quercus petraea* trees were identified and further grouped into eight distinct populations of different sizes based on their spatial clustering (Figure 1, Table S3). Within three populations, the co-occurrence of other oak species, namely *Q. robur* and *Q. cerris*, was observed. *Q. robur* was found to coexist in two populations, specifically in Fugnitz 1 (FG1) and Fugnitz 2 (FG2). *Quercus cerris* was observed in the extremely dry part of Kirchenwald population. Both oak species, *Q. robur* and *Q. cerris*, do not intermingle with the *Q. petraea* individuals in the study area; there is rather a transition between species when the site conditions change. The age of the sampled mother trees was determined through a comparative core analysis. The mean age of the sampled trees in the Thayatal National Park was 132 years in 2023, ranging from 29 to 245 years (Figure 1). Within each population, a distinct pattern of age distribution was observed (Figure S3, Supplementary Materials S3).

### 3.1. Autochthony

In this study, we employed reference samples for six distinct haplotypes obtained through PCR-RFLP analyses by [25,35–37,39] (Figure 2A). Nevertheless, we identified only one combination of cpSSRs fragment lengths for all sessile oak samples from the Thayatal National Park, grouping in a cladogram with reference samples of PCR-RFLP haplotype 17a. The haplotype 17a was previously identified in neighboring populations around the National Park (Figure 2B). Notably, the PCR-RFLP haplotypes 5, 6, and 7 displayed an identical combination of allelic variants (Table S3, Supplementary Materials S3), rendering

them indistinguishable using the set of microsatellite cpDNA loci utilized in this research. The markers ccmp6 and μkk3 were not variable, but the rest demonstrated two to four different alleles with 1bp difference, with μdt4 and μcd4 being the most diverse and having three and four alleles, respectively (Table S3).

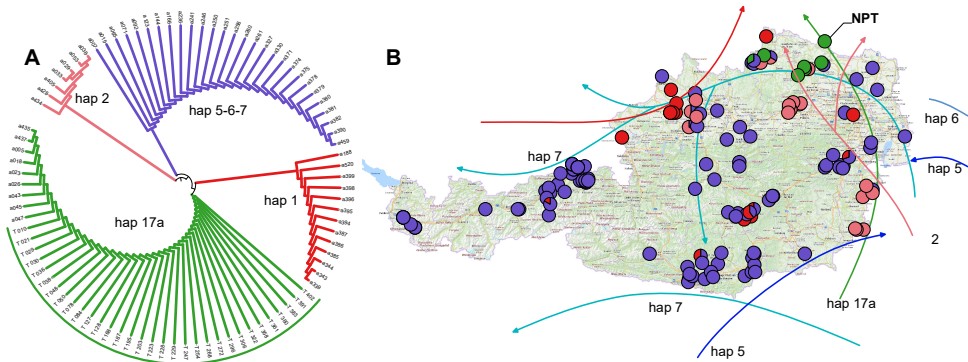

**Figure 2.** Thayatal sample clustering and haplotype distribution in Austria. (**A**) Neighbor-joining dendrogram with samples from the Thayatal National Park (labeled as T and located along the green arc) and selected reference haplotypes from [39]. (**B**) Distribution of different cpDNA haplotypes [39], together with recolonization routes from [26,35–37]. The population in the Thayatal National Park is labeled as NPT. The colors are adopted from [26] with modifications. Since cpSSRs do not distinguish between haplotypes 5, 6, and 7, the color of these haplotypes is intermediate between the individual original colors defined by [36].

### 3.2. Genetic Diversity and Genetic Structure

Given the variation in population sizes, ranging from 32 to 83 individuals, our focus here was on comparing patterns of genetic diversity using two standardized diversity metrics: unbiased expected heterozygosity ($Hs$) and allelic richness ($Ar$), which allow such comparisons between populations (Table S4, Supplementary Materials S3). The observed $Hs$ values exhibit a trend, with the highest values (ranging from 0.87 to 0.88) observed in populations situated in the north-western (SF, HK, MP, FG1, FG2) and central (UB) parts of the study area. Conversely, the lowest $Hs$ values are apparent in two populations (KW1 and KW2) located in the south-eastern region. Notably, allelic richness ($Ar$) displays a similar pattern. Populations in the north-western and central parts (except population FG2 with an $Ar$ value of 10.70) demonstrate higher richness values, ranging between 11.66 and 12.73, in comparison to populations KW1 and KW2, which exhibit $Ar$ values of 10.6 and 9.0, respectively. The analysis of the inbreeding coefficient ($G_{IS}$) revealed that, except for the population KW1, all populations in the National Park exhibit a significant excess of homozygotes, whereas KW2 demonstrates an insignificant deviation from Hardy-Weinberg equilibrium.

The analyses of pairwise differentiation, as determined by Jost's D measure that quantifies the real relative extent of differentiation using allele frequencies within populations, revealed the most pronounced differentiation for KW2. This population exhibited consistently elevated differentiation values (ranging from 0.092 to 0.1542) in comparison to all other populations except its immediate neighbor KW1. Subsequent to KW2, KW1 also demonstrated notable differentiation values (ranging from 0.0685 to 0.1278) in relation to all other populations except KW2. The majority of KW1 and 2/3 of trees from KW2, located in the eastern part of the National Park, differ from the other oak populations in the Thayatal National Park (Figure 3), as revealed by analysis of the genetic structure using the software STRUCTURE (Figure 3A) and PolyRelatedness (Figure S4, Supplementary Materials S3) clustering the individuals into two groups based on their multilocus genotypes (Figure 3A,B). Once the individuals of the smaller cluster (KW1 and the majority of KW2) were removed, it was no longer possible to identify any genetic structure (Figure 3C). Consequently, these individuals (in total 96) were excluded from the analyses of drought tolerance.

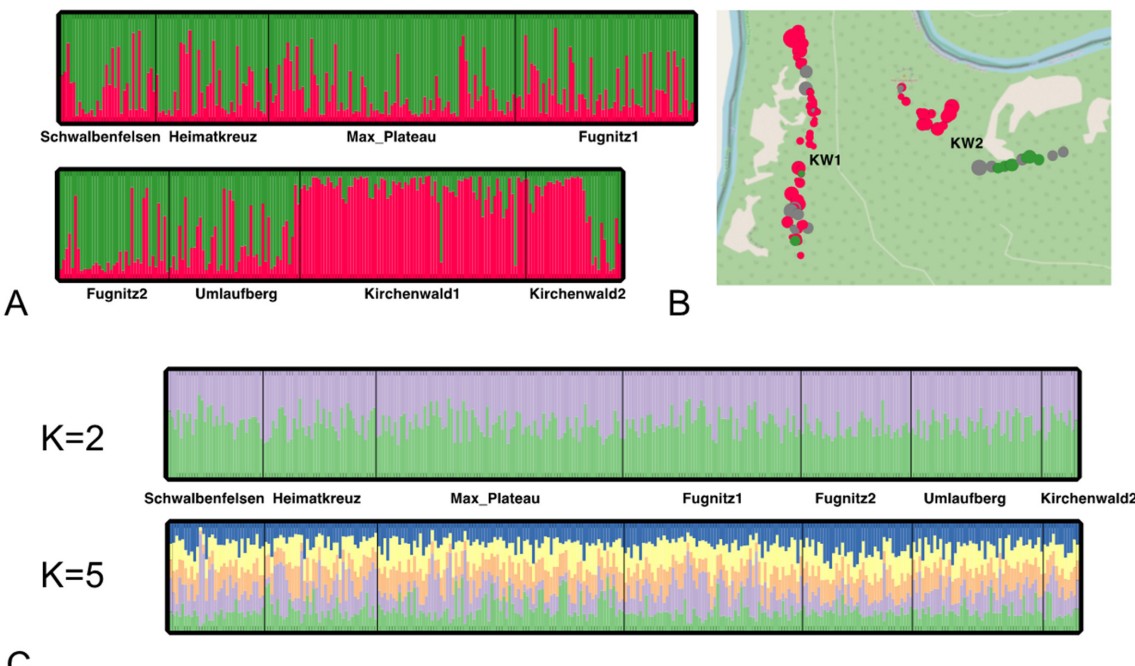

**Figure 3.** Genetic structure in *Q. petraea*. (**A**) Individual membership proportions determined by a STRUCTURE analysis across all populations. Each individual is represented with a vertical bar, and inferred clusters are marked with a different color. (**B**) Map with individuals from Kirchenwald1 and Kirchenwald2. The diameter of points represents the age of each tree as of 2023. Points with Q > 0.75 are colored red, Q < 0.25—green, grey—none of those. (**C**) Individual membership proportions excluding Kirchenwald1 and part of Kirchenwald2 individuals. Results are presented for K = 2, allowing comparison to 3(**A**), and K = 5 as suggested by the STRUCTURE analysis.

### 3.3. Drought Tolerance

Selection of the most extreme phenotypes per trait (*Rt*, *Rc*, *Rs*, r*Rs*) resulted in two subgroups "good" and "bad" that significantly differed for all three drought events studied (1917,1947,1992–1994, Figure 4). In total, 75 trees were selected for *Rt* (Good *Rt* = 36, Bad *Rt* = 39), 69 for *Rc* (Good *Rc* = 38, Bad *Rc* = 31), 63 for *Rs* (Good *Rs* = 29, Bad *Rs* = 34), and 68 for r*Rs* (Good r*Rs* = 37, Bad r*Rs* = 31). The best-performing trees for each trait were combined, resulting in a set of 85 selected trees representing potentially drought-tolerant phenotypes (Figure 5, Figure S5, Supplementary Materials S4 [72]).

Furthermore, based on the isotopic analysis of the $\delta^{13}$C of latewood in wet (1987) and dry (1994) years, it was found that the estimated iWUE did not show significant differences among groupings of selected trees (good/bad) for all four traits (*Rt*, *Rc*, *Rs*, r*Rs*, Figure S6A–D, Supplementary Materials S4). In addition, DWiWUE (equal to dry iWUE minus wet iWUE) showed no significant differences between the groupings for any of the selections made (Figure S6E). Finally, three of the four traits (*Rt*, *Rc*, r*Rs*) were significantly repeatable across drought events: Recovery showed the highest degree of repeatability across years (*Rc*; R = 0.088), followed by resistance (*Rt*; R = 0.082) and relative resilience (r*Rs*; R = 0.065), while resilience showed no repeatability at all (*Rs*; R = 0.0). Estimates of all repeatability values and their confidence intervals are given in Table S6 (Supplementary Materials S4), and the uncertainty quantified by parametric bootstrapping was plotted (Figure S7, Supplementary Materials S4). Our study confirms the existence of genetic variation in drought response among individuals within the national park. The variations in the levels of repeatability observed can be attributed to the complexity of physiological processes that underlie the various dendrochronologically-based drought measures [73].

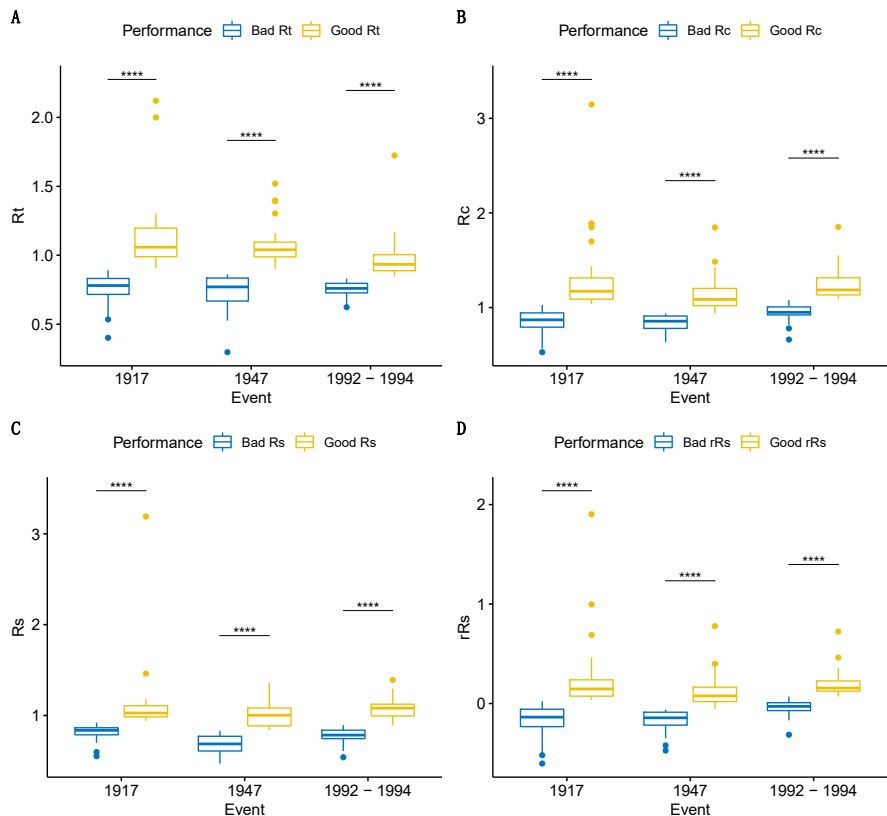

**Figure 4.** Box plots for selected groups ("good" and "bad"; see explanation in paragraph 2.4) for each trait across all drought events. (**A**) Resistance. (**B**) Recovery. (**C**) Resilience. (**D**) Relative resilience. All pairwise *t*-test comparisons showed significant differences (**** *p*-value $\leq$ 0.0001).

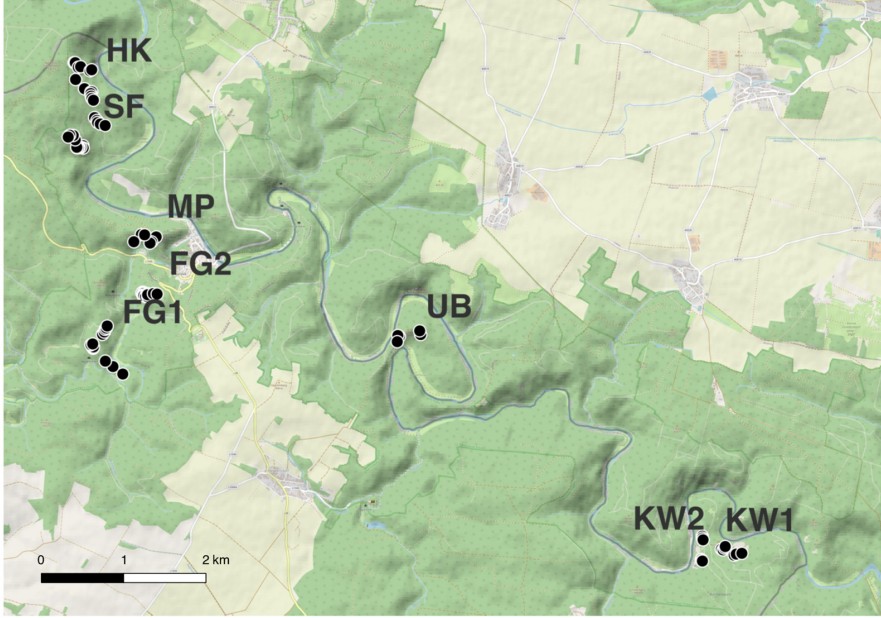

**Figure 5.** Distribution of all 85 selected and potentially drought-tolerant trees (black dots with white outline) across the Thayatal National Park.

Analyses of climatic trends in the Thayatal National Park, as displayed by alternations in mean annual precipitation and mean temperature, revealed a sharp temperature increase after 1960 (Figure S8A, Supplementary Materials S4) while indicating a slight decline in annual precipitation from 1960 to 2020 (Figure S8A). More specifically, the mean annual

temperature increased from 8.3 °C between 1900 and 1960 to 10.3 °C (1960 to 2020), resulting in a difference of 2 °C. A similar but opposite trend was observed for the precipitation: The mean annual precipitation decreased from 538 mm (1900–1960) to 490 mm (1960–2020), denoting a 9% reduction compared to the preceding period.

Our analysis of correlation revealed that the age only had an impact on the responses observed for the iWUE of the wet year and the difference between dry and wet years (DWiWUE) but not on the growth-dependent variables (Figure S8, Supplementary Materials S4). iWUE variables were moderately correlated with each other (0.57) but showed no correlation at all with the growth-dependent variables, which showed moderate and high correlations with each other.

## 4. Discussion

### 4.1. Autochthony

Highlighted by an identical cpDNA pattern derived from cpSSRS and aligned within the context of cpDNA polymorphism at the regional level, the studied *Q. petraea* within the Thayatal National Park demonstrates an autochthonous origin. In more detail, the 17a haplotype displayed an exclusive presence across the eight studied populations in the Thayatal National Park. This particular haplotype has also been identified in *Q. robur* within the Thayatal National Park (S. Jansen, personal communication) and white oak populations in the adjacent regions [35,38] (Figure 2), including a large population close to the village of Sigmundsherberg situated 20 km away. This population encompasses a remarkable 395 trees exceeding 100 years in age, constituting representatives of *Q. robur*, *Q. petraea*, and their hybrids [39].

The age distribution of trees within the park spans from 29 to 245 years for the entire *Q. petraea* cohort (as of 2023; Figure 1). Furthermore, distinctive age patterns are evident among various populations, with a notable minimum difference of 100 years between the youngest and most mature *Q. petraea* specimens (Figure S3, Supplementary Materials S3). These observations collectively suggest that the studied trees, primarily occupying rocky slopes and dry grassland sites, underwent a process of natural regeneration.

Nonetheless, it remains crucial to bear in mind that the extent of human intervention in Central European forests has been so pronounced that one cannot accurately reference them as original sensu virgin forests anymore [74]. This prompts a brief overview of human influence within north-eastern Austria and neighboring regions. The historical trajectory of oak stands in this region, north-east Austria, and in the valley of the Danube commences around 10,000 years BP [75,76]. An exploration of archaeological findings reveals the existence of humans in Austria and the adjacent Czech and Slovak Republics during the Middle Paleolithic Era (300,000–50,000 BP [77]). However, limited by small populations and rudimentary tools, their impact on nature was not substantial. Anthropogenic influence on natural vegetation began as early as 7000 BP in central Europe [75] during the Neolithic Revolution, marked by deforestation for agricultural purposes. By about 3000 BP, the forests had undergone substantial transformations, rendering natural regeneration insufficient [74]. Different cultures and tribes have exerted varying degrees of influence on forests. For instance, the Celts and Romans played a significant role in forest degradation, as the former cultivated land, raised animals, and used wood for metallurgy, while the latter exploited forests for commercial timber export. The arrival of the Western Slavs in the 5th century initially led to reduced exploitation owing to their beliefs, but this dynamic shifted with the advent of Christianity. In the 13th century, Mongol invaders exacerbated deforestation by burning villages and forests. Subsequently, the need for extensive oak resources emerged for pit timber, charcoal for smelting, and lime production. Large quantities of wood were transported via artificial streams and rivers, especially Danube floats downstream to Vienna and Budapest, resulting in an acute wood scarcity. A significant contribution to forest renewal occurred with the enactment of Maria Theresia's Statute in 1769, with most of its forest regulations, subject to minor modifications, continuing to guide reforestation and silviculture practices. Consequently, substantial transformations have transpired since

the 19th and 20th centuries due to artificial reforestation, primarily involving spruce and also extending to oaks. Forest grazing played a major role from the 17th to the 19th century. In Hardegg (in the study area) during this time, many clothiers' businesses processed sheep wool. Historical reports document discussions about grazing rights in the church forest (KW 1 and KW 2) [78].

Austria has been revealed to be a crossroad, a meeting point for various post-glacial migration routes of white oaks. The country has been colonized by six distinct haplotypes originating from Italy and the Balkans (Figure 2B). The haplotype 17a, which has been identified in the Thayatal National Park, represents just one among several variants within the 17 group (17a–f). This haplotype is proposed to migrate through the northern Adriatic region before spreading eastward into Croatia, Hungary, and Austria [26,79]. Among all the variants of haplotype 17, it stands out as the most common and exhibits a more or less continuous distribution along the Apennine chain in the Italian Peninsula. This peninsula played a multifaceted phylogeographic role, hosting numerous glacial refugia and serving as a significant biogeographical threshold [28].

### 4.2. Patterns of Genetic Structure and Diversity Linked with Age Structure

The study of the genetic diversity, spatial structure, and differentiation patterns of *Q. petraea* trees within the national park has revealed a distinct partitioning. Specifically, trees in the north-western and central areas differ from those in the north-eastern area. Notably, the former group exhibits increased genetic diversity ($Hs$, $Ar$) compared to the latter. In more detail, the KW1 population is the most distinctive, characterized by the lowest values for both diversity indices ($Hs$ = 0.759, $Ar$ = 9.011, Table S4, Supplementary Materials S3). This distinction is further evident in membership proportions identified through STRUCTURE analyses (Figure 3), in contrast to the remaining populations ($Hs$ = 0.823–0.89, $Ar$ = 10.60–12.43). The KW2 population, located closest to KW1, occupies an intermediary position in terms of diversity indices and genetic structure. Around two-thirds of individuals (as determined by membership proportions from STRUCTURE), which cluster geographically together, resemble KW1's trees, while the remaining portion aligns with the other populations (Figure 3B). The presence of spatial structure within *Q. petraea*, encompassing a relatively small area with a maximum separation of approximately 10 km between the most distant populations, is surprising. This is noteworthy as *Q. petraea* is conventionally recognized for its genetic homogeneity, maintained even over extensive geographic distances and habitat fragmentation (see [80], along with cited examples). However, when considering the age structure of the studied populations (which are grouped based on their geographical position), it becomes evident that individuals within KW1, and the subset of KW2 resembling KW1, comprise a cohort of young trees. We hypothesize that this pronounced structure revealed largely in KW1 and part of KW2 can be attributed to the prevalence of young age cohorts (around 60 years old, Figure S3, Supplementary Materials S3) in them. These young trees of KW2 also exhibit spatial clustering (Figure 3B). The influence of age on genetic structure has been documented in other studies. Specifically, a notable fine-scale genetic structure was detected in the youngest cohort, while such patterns were absent in older cohorts (as demonstrated in *Populus nigra* [81]). Additionally, changes in genetic structure might arise due to variations in tree density, with higher densities observed in young trees and lower densities in older trees (as exemplified in [82]) and clear-cut.

### 4.3. Response to Extreme Drought Years and Selection of the Best Drought-Adapted Phenotypes

The tree response to three historical drought events (1992–1994, 1947, and 1917) covered a span of 75 years within a tree's lifetime. It was primarily this broad time span between the selected drought events and, to a smaller extent, the exclusion of samples from KW1 and some from KW2 due to their discernible pattern in the genetic structure that led to a reduction in our initial group of 404 trees to 253 (37%). However, our aim was to identify relevant drought-tolerant trees, so their exposure to all tree drought events

was essential to obtain robust patterns in drought response. Consequently, we focused our analysis on this reduced set of trees in this part of our study. Each individual drought event exhibited distinct characteristics. Specifically, the year 1917 exhibited the most severe deficiency in precipitation sum during the vegetation period (May to September) among all 125 time series documented by the climatic station. In contrast, the year 1947, along with its extension into 1948, witnessed low precipitation following an extended period (~1935–1945) characterized by favorable precipitation conditions during the growing season (May to September). Furthermore, 1992 to 1994 were marked by elevated temperatures and low precipitation. Beyond the specifics of the drought events, the trees within the national park were exposed to overarching climate trends, with a notable increase in mean temperature coupled with a reduction in annual precipitation. This multifaceted assessment sheds light on the intricate interplay between historical drought occurrences and broader climatic shifts, deepening our understanding of tree responses in the context of evolving environmental conditions.

To identify the best drought-adapted phenotypes, we examined five traits, including four morphological traits ($Rt$, $Rc$, $Rs$, $rRs$) and one physiological trait, iWUE (water use efficiency), measured by $\delta^{13}$ C [83]. Our goal was to strengthen the robustness of the selection process by considering all these traits together. However, we found no significant overlap when we compared the phenotypes identified as "good" or "bad" based on the morphological measures with those identified using iWUE. Because of this lack of alignment, we have decided not to include iWUE in the selection of the potentially drought-tolerant phenotypes. This discrepancy may be due, in part, to the tree ring growth response, which is not only governed by carbon uptake via photosynthesis but also carbon allocation [9–11] and the influence of nutrient availability on iWUE [84]. In addition, photosynthesis and stomatal conductance can both vary and not always in the same way [23].

When delving into the responses of our groups ("good" or "bad") over the three drought periods and comparing them (Figure 4), we observed that the most resistant trees ("good" $Rt$), those that can effectively buffer drought stress ($Rt = 1$ indicates complete resistance), are gradually approaching the vulnerability of drought-susceptible trees ("bad" $Rt$), especially in the years 1992–1994. Evidence already exists indicating a decline in *Q. petraea* populations at the dry boundary of its range in the White Carpathian wooded grasslands located in the Czech Republic. This decline is particularly notable in areas exposed to significant insolation, where severe droughts have been recurring during the past three decades [85]. Focusing on $Rc$, a trait that characterizes a tree's ability to recover growth after stress, where $Rc = 1$ suggests the persistence of pre-drought growth levels, $Rc < 1$ indicates further decline, and $Rc > 1$ signifies recovery from the growth levels experienced during drought. Analyzing the response of the group with "bad" $Rc$, during the years 1992–1994 marked by not only low precipitation as in 1917 and 1947 but also elevated temperatures, "bad" $Rc$ trees exhibited better performance compared to earlier events, possibly due to the significant growth reduction observed in 1992–1994. Regarding $Rs$, it appears that the 1947 event had a more pronounced impact on resilience, particularly for the less resilient trees. The resilience index $Rs$ measures the ratio between average growth after a stress event and before the drought event, thereby assessing a tree's capacity to return to pre-drought growth rates; $Rs \geq 1$ indicates a full recovery or increased growth after the drought event, while $Rs < 1$ indicates a decline in growth. Further detailed analyses could shed light on whether the post-drought conditions were conducive to explaining these patterns effectively. Considering the current drought stress event growth for the relative resilience calculation, it again seems that 1992–1994 was the most severe event, as "bad" $rRs$ trees are in a better performance compared to the previous events (due to the drastic growth reduction in 1992–1994 and the subsequent good recovery).

When considering the results of all four studied traits, it becomes evident that the group of "good" trees among the four measures does not always consist of the same individuals. It is crucial to carefully weigh the trade-offs that occur when selected individuals show poor performance in another trait. Specifically, 19 individuals exhibited trade-offs,

leaving only 66 out of 85 without any trade-off. When selecting for resistance ("good" *Rt*) and resilience ("good" *Rs*) simultaneously, this number narrows down to just six trees: T102, T225, T245, T271, T027, and T046. This raises the question of the primary or more critical strategy for coping with drought. Is it better to keep growing during drought stress (*Rt*) or temporarily halt growth during drought and then exhibit substantial recovery (*Rc*, *Rs*)? One option is to focus on the results of resistance since it directly captures a tree's response to ongoing drought and is, therefore, the most direct parameter of drought response [73]. The other indices involving the recovery period are likely influenced by post-drought climate conditions or the individual allocation of non-structural carbohydrates (NSC). Further investigation (both under controlled conditions and in the field) was initiated to delve into these phenotypes in greater detail.

## 5. Conclusions and Outlook Related to Breeding and Conservation

In recent decades, the risk of drought stress has escalated in many regions due to climate change, which is characterized by rising temperatures, altered precipitation patterns, and accelerated snowmelt (IPCC Sixth Assessment Report). Drought stress in deciduous formations, such as temperate forests, can also arise due to gaps in the tree or shrub canopy. These gaps cause increased evapotranspiration, elevated exposure to sunlight, and, consequently, a decrease in water and humidity within the microclimatic environment [86]. Therefore, understanding plant responses and adaptation mechanisms to drought and identifying phenotypes with heightened drought tolerance has become increasingly important.

Our study examined the autochthony, genetic diversity, and structure of *Q. petraea* populations in the ecologically diverse Thayatal National Park to provide a basis for the identification and selection of drought-adapted phenotypes. The presence of a consistent haplotype (17a) across *Q. petraea* populations in the national park, along with its identification in neighboring regions, underscores the longstanding and local history of these oak populations. This finding aligns with the notion that these trees have thrived in the region over an extended period. Our analyses shed light on the existence of genetic structure among sessile oaks within the park, which is reflected in the observed patterns of genetic diversity. Notably, populations in the north-western and central regions displayed higher genetic diversity, while those in the south-eastern region exhibited lower diversity values. The distinctive genetic makeup of KW2 and KW1 further emphasizes this genetic structure, leading to the exclusion of their individuals from the selection of potentially drought-tolerant trees. To assess the drought responses of the final set of 253 trees, we employed both morphological and physiological measures recorded in situ within tree rings. Specifically, we used growth-based indices of drought response (as described by [41]) and intrinsic water use efficiency (iWUE) evaluations. Our analysis of responses to three distinct drought episodes allowed us to identify a subset of 85 potentially drought-tolerant trees. These selected trees and their progeny form the foundation for subsequent breeding and conservation efforts. More specifically, these trees are essential for (a) confirming the observed drought responses under controlled conditions, facilitated by High-Throughput Plant Phenotyping (HTPP) and RNA-Seq analyses, and (b) initiating trials using the selected phenotypes to meet the increasing demand for climate-adapted FRM.

To enhance our understanding of the selected 85 phenotypes from this study, we are further evaluating the progeny of these trees and their drought responses by using HTPP. This is planned to be followed by a whole-transcriptome approach, RNA-seq, to link phenotypes to genotype-specific expression markers. Newly emerging technologies, such as HTPP, combined with more established high-throughput genotyping, offer significant improvements in terms of data quality, time efficiency, spatial coverage, and cost-effectiveness [87,88].

From a forestry perspective, obtaining seeds and planting material from suitable drought-tolerant sessile oak populations is crucial, given that the supply of such plant material in Austria is not available. From a conservation perspective, it is equally important to preserve forest genetic resources. To address these needs, a new 1.3-hectare plot was

established in the spring of 2023. This plot comprises the progeny of selected drought-tolerant trees and is intended to contribute to the long-term seed supply for dry and challenging ecological sites, benefiting both nature conservation and forestry.

Drought tolerance is a complex trait that can be achieved through many strategies [89]. Recognizing that these adaptation strategies are limited depending on the tree species' constraints is, however, essential.

**Supplementary Materials:** The following supporting information can be downloaded at: https://www.mdpi.com/article/10.3390/f14112225/s1, Figure S1a: Selected drought years and the SPEI-6 time series; Figure S1b: Trends in mean annual precipitation, mean seasonal precipitation, and mean temperature estimated for the period 1900–2020 derived from the climatic data measured at station Retz. Figure S2: Histogram with normal distribution curve (red line) and Q-Q plot for all traits on all studied drought events (A) Resistance. (B) Recovery. (C) Resilience. (D) Natural logarithm of resilience. (E) Relative resilience. (F) Lilliefors test results for all traits and drought events, green highlighted those that significantly ($p$-value $\geq 0.01$) fit a normal distribution. Figure S3: Histograms of tree ages (year 2023) for each location under study with age at the x-axis and frequency at the y-axis. Figure S4: Heatmap plot of genetic relatedness matrix calculated in PolyRelatedness. Figure S5: Venn diagrams for selected genotypes in both (A) "good" and (B) "bad" performance on the traits under study. Figure S6: Box plots of intrinsic water use efficiency (iWUE) for selected groups (good and bad) for each trait, when isotopic analysis of $\delta 13C$ of latewood was performed: wet (1987) and dry (1994) years. (A) Resistance. (B) Recovery. (C) Resilience. (D) Relative resilience. (E) Difference between dry and wet values (DWiWUE) box plots for all traits and groups. All pairwise $t$-test comparisons showed no significant differences ($p$-value $> 0.05 -$ ns). Figure S7: Repeatability estimates for each trait ($Rt$, $Rc$, $Rs$, Ln($Rs$), r$Rs$). Distribution of the parametric bootstrap samples along with the point estimate and the limits of the confidence interval. Figure S8: Correlogram based on the correlation matrix (Spearman) of all studied traits through all studied drought events (1917, 1947, 1992–1994) and corresponding age. The correlation matrix was ordered according to the degree of association between variables (hierarchical clustering) and the significance threshold was set at $p$-value $\leq 0.01$. Only significant correlation coefficients are plotted, sized and colored according to their value. Table S1: Summary of genotype loci including locus name, type, primer sequences, fluorescent dyes, multiplex combinations (including concentration in the multiplex of each primer), measured fragment sizes, and sources of primers. Table S2: Reference samples with different haplotypes revealed by PCR-RFLP analysis (Petit et al. [22] and Tutková van Loo & Burg [39]). Table S3: Genotypes of haplotypes revealed by PCR-RFLP analysis Petit et al. (2002) in the present analysis with 10 cpSSR markers. Table S4: Measures of genetic diversity within populations based on nuclear SSR. Table S5. Results of pairwise values of D$_{JOST}$ (2008) calculated in R package diveRsity (Keenan et al. [58]) with associated confidence interval in [90]. Table S6: Significant repeatability estimates ($p$-value $\leq 0.05$) for each studied trait including the standard error and 95% confidence interval.

**Author Contributions:** Conceptualization, M.v.L.; methodology R.U., S.P., C.T.-M., M.v.L., A.W., M.G. and F.I.; software and analyses R.U. and C.T.-M.; writing—original draft preparation, M.v.L., C.T.-M., R.U., A.W. and C.Ü.; writing—review and editing all authors; supervision, M.v.L.; funding acquisition, H.K. All authors have read and agreed to the published version of the manuscript.

**Funding:** The creation of the analyzed data was supported by the Austrian federal government, the federal provinces, and the European Union (project: TERZ, grant number 8.5.2-III4-06/19, LE 14-20).

**Data Availability Statement:** Data is contained within the article or Supplementary Materials or at DRYAD (https://doi.org/10.5061/dryad.6djh9w16k) (accessed on 22 October 2023).

**Acknowledgments:** We would like to acknowledge Dominik Lorenschitz, Michael Kober-Eberhardt, Carla Maria Schengili, Arnold Triebelnig, and foresters and rangers of the Thayatal National Park for their support.

**Conflicts of Interest:** The authors declare no conflict of interest. The funders had no role in the design of the study; in the collection, analyses, or interpretation of data; in the writing of the manuscript, or in the decision to publish the results.

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
