# Peer review of "Quercus petraea (Matt.) Liebl. from the Thayatal National Park in Austria: Selection of Potentially Drought-Tolerant Phenotypes"

_forests, doi:10.3390/f14112225_

Round 1
Reviewer 1 Report
Comments and Suggestions for Authors
The study by Marcela van Loo and collaborators entitled “Identification and conservation of valuable forest genetic resources of Quercus petraea from the Thayatal National Park in Austria: selection of drought tolerant phenotypes and assessment of autochthony and genetic diversity” aims to detect climate-adapted forest germplasm to guide tree breeding and conservation strategies.
1. The manuscript has clear objectives but fails to combine both approaches, i.e. the genetic information using neutral molecular markers to analyze autochthony and diversity with phenotypic characters by dendrochronological techniques and ecophysiological traits of sensitivity to drought, to detect elite trees. I am aware of the limitations to relate both methodologies given that the molecular markers used are neutral. Unless genomic techniques as SNPs that consist of adaptive and neutral signals are applied, a difficulty exist in developing genotype-phenotype analyses (Fasanella et al 2021). Nonetheless, it seems to me that an effort to combine the two data sources is missing. For example, what is the genetic profile of trees considered good and bad in terms of dendrophenotypes? Do the “good” trees have particular alleles / genetic diversity levels?
Fasanella, M.L. Suarez, R. Hasbún, and A.C. Premoli. 2021. Individual-based dendrogenomic analysis of forest dieback driven by extreme droughts. Canadian Journal of Forest Research. 51(3): 420-432. https://doi.org/10.1139/cjfr-2020-0221
2. Data analysis of water use efficiency seems to have been unexplored. Authors have analyzed WUE only in relation to bad/good dendrophenotype traits. No attempt has been made in combining WUE with genetic information of individual genotypes/alleles as stated above. Similarly, no geographic information is provided on the possible location of potentially elite phenotypes and to relate this information with the authoctony/genetic diversity of genotypes.
3. The analysis of authoctony as well as reference to possible migration routes during glacial eras is outdated. Numerous pieces of evidence exist in this regard and should be included in the introduction and in relation to the discussion of autochthony of germplasm.
Bhagwat, Shonil A., and Katherine J. Willis. “Species Persistence in Northerly Glacial Refugia of Europe: A Matter of Chance or Biogeographical Traits?” Journal of Biogeography, vol. 35, no. 3, 2008, pp. 464–82. JSTOR, http://www.jstor.org/stable/30054708. Accessed 27 Sept. 2023.
Schmitt, T., Varga, Z. Extra-Mediterranean refugia: The rule and not the exception?. Front Zool 9, 22 (2012). https://doi.org/10.1186/1742-9994-9-22
Parducci L et al. 2012 Glacial survival of boreal trees in northern Scandinavia. Science 335, 1083 – 1086. (doi:10.1126/science.1216043)
Svenning JC, Normand S, Kageyama M. 2008 Glacial refugia of temperate trees in Europe: insights from species distribution modelling. J. Ecol. 96, 1117– 1127. (doi:10.1111/j.1365-2745.2008.01422.x)
4. Also in the same line with previous comment, recolonization routes shown in Figure 2B should be revised and Figure 2A should be thoroughly discussed. Haplotypes exclusively present in Thayatal National Park, which demonstrate an autochthonous origin, are only briefly discussed. Panel A clearly shows that haplotype 17a is closely related to haplotypes 1 and 2 which are geographically restricted, the latter of which with shows a clear early divergence. These three haplotypes may well suggest long-lasting presence in the area and not recolonization from further away sources which need to be taken into consideration.
Other comments
Tittle is too long
L106, References [25,36] are old, some new studies suggest refugia further north and towards the east in continental Europe and not just in southern peninsulas
Reference 36 is incomplete
Legend Figure 2 not clear where the circumference where the Thayatal National Park is located, use another symbol
Tables S2 and S4, latitude and longitude of populations are confounded
Figure 1 use bigger font for populations’ names
Table S4. Measures of genetic diversity are these nuclear SSRs? Please clarify
Author Response
Dear Reviewer 1. Thank you very much for taking the time to review our manuscript. In the attached document you will find our detailed response to all your suggestions and comments. In the attached manuscript and the supplementary material we also consolidated changes suggested by the other two reviewers.
Please see the attachment

Reviewer 2 Report
Comments and Suggestions for Authors
I an not able of giving a veredict concerning this paper, in spite of the ideas presented in the Introduction and the good use of the methods. The paper deals with TOO many questions that are mixed along the text, with no clear conclusions about any of them.
I can only suggest to re-write the paper, focussing in one or two (e.g. genetic diversity plus autochtony) or water relations and drought event responses. In fact, the information given might be enough for writting two papers with an adequate splitting of the data. In its present form, it is really hard to extract a significative information about the performance of this oak in the study area.
As the paper is quite long, I encourage the authors to i) reduce the information given, ii) select and expose the most brillant results they consider.
Author Response
Dear reviewer 2,
Thank you very much for taking the time to read our manuscript. At the end of this document, you will find our responses to all your suggestions and comments. In the attached manuscript and the supplementary material, we also consolidated changes suggested by the other two reviewers.
Please see the attachment.

Reviewer 3 Report
Comments and Suggestions for Authors
Congratulations on the presentation of this manuscript, which is very interesting and important for understanding the effects of accelerated global warming on deciduous tree species typical of temperate climates such as Quercus petraea or other Quercus sp. The paper is well written, except for minor conceptual issues. The text is easy to read and the supplementary material shows the authors' effort to make their manuscript understandable.
To see the comments, please see the attached document

The English seems adequate, but there are some expressions that are not well understood such as "elite" or "mother tree".
Author Response
Dear Reviewer 3, Thank you for dedicating your time to review our manuscript and for providing us with your valuable comments. We have incorporated the majority of your suggestions into the manuscript and the supplementary material. You can locate these changes directly in the revised manuscript. We have also addressed your comments individually below. Furthermore, we have removed the terms "elite" and "mother tree" as they were unnecessary.
"Please see the attachment."

Round 2
Reviewer 1 Report
Comments and Suggestions for Authors
Authors have succesfully responded to all comments
One minor suggestion: Probably the title could be
Forest genetic resources of Quercus petraea (Matt.) Liebl. from Thayatal National Park, Austria: selection of potentially drought tolerant phenotypes
Reviewer 2 Report
Comments and Suggestions for Authors
Authors have made a great effort to include all the suggestions given by independent reviewers and the overall paper quality has improved. I am still concern with the amount of information given in this paper, that can be a kind of confussing. Nevertheless, I have found that so much information can be of interest for other researchers when searching for global trends in such an important species.
So, I will recommend to accept the manuscript in its present form, as the presentation is absolutely right in my opinion.